# Use of Impedance Spectroscopy for the Characterization of In-Vitro Osteoblast Cell Response in Porous Titanium Bone Implants

**Mercè Giner** [1,*], **Alberto Olmo** [2,3,*], **Miguel Hernández** [2,3], **Paloma Trueba** [4], **Ernesto Chicardi** [4], **Ana Civantos** [5], **María Ángeles Vázquez** [6], **María-José Montoya-García** [6] and **Yadir Torres** [4]

1 Departamento de Citología e Histología Normal y Patológica, Universidad de Sevilla, Avda. Doctor Fedriani s/n, 41009 Sevilla, Spain

2 Instituto de Microelectrónica de Sevilla, IMSE-CNM (CSIC, Universidad de Sevilla), Av. Américo Vespucio s/n, 41092 Sevilla, Spain; miguelhernandezcamacho@gmail.com

3 Escuela Técnica Superior de Ingeniería Informática, Departamento de Tecnología Electrónica, Universidad de Sevilla, Avda. Reina Mercedes s/n, 41012 Sevilla, Spain

4 Departamento de Ingeniería y Ciencia de los Materiales y del Transporte, Escuela Politécnica Superior, Calle Virgen de África 7, 41011 Sevilla, Spain; ptrueba@us.es (P.T.); echicardi@us.es (E.C.); ytorres@us.es (Y.T.)

5 Department of Nuclear, Plasma and Radiological Engineering, College of Engineering, University of Illinois at Urbana-Champaign, Urbana, IL 61801, USA; ancife@illinois.edu

6 Medicine Department, University of Seville, Avda. Dr. Fedriani s/n, 41009 Sevilla, Spain; mavazquez@us.es (M.Á.V.); pmontoya@us.es (M.-J.M.-G.)

* Correspondence: mginer@us.es (M.G.); aolmo@dte.us.es (A.O.); Tel.: +34-627-21-7746 (M.G.); +34-09-5455-4325 (A.O.)

**Abstract:** The use of titanium implants with adequate porosity (content, size and morphology) could solve the stress shielding limitations that occur in conventional titanium implants. Experiments to assess the cellular response (adhesion, proliferation and differentiation of osteoblasts) on implants are expensive, time-consuming and delicate. In this work, we propose the use of impedance spectroscopy to evaluate the growth of osteoblasts on porous titanium implants. Osteoblasts cells were cultured on fully-dense and 40 vol.% porous discs with two ranges of pore size (100–200 μm and 355–500 μm) to study cell viability, proliferation, differentiation (Alkaline phosphatase activity) and cell morphology. The porous substrates 40 vol.% (100–200 μm) showed improved osseointegration response as achieved more than 80% of cell viability and higher levels of Cell Differentiation by Alkaline Phosphatase (ALP) at 21 days. This cell behavior was further evaluated observing an increase in the impedance modulus for all study conditions when cells were attached. However, impedance levels were higher on fully-dense due to its surface properties (flat surface) than porous substrates (flat and pore walls). Surface parameters play an important role on the global measured impedance. Impedance is useful for characterizing cell cultures in different sample types.

**Keywords:** porous titanium; electrical impedance spectroscopy; cell differentiation; surface roughness

## 1. Introduction

Studies carried out in Spain indicate that in 2050 more than 1/3 of the inhabitants will be 65 years of age or older [1]. The disorders and deterioration of the skeletal-muscle system affect the quality life of patients and generate high costs to the healthcare system. However, in addition to the loss of bone mass and mechanical resistance of the tissues with age, we should take into account traumas and

congenital or degenerative diseases [2]. For example, diseases such as osteoarthritis, spondylosis and dental caries cause irreversible structural damage to the affected bone tissues that require therapies for repair. When non-invasive therapies do not respond, surgical intervention is the next option, which involves removing the damaged tissue and replacing it, often, with a synthetic implant to restore function and reduce pain. In this context, the search for innovative solutions to improve health care for the population of this stage of life remains a global challenge. Among the materials used in orthopedics with these fines, titanium and its alloys play an important role, due to their excellent mechanical and corrosion resistance and their adequate biological behavior [3–10]. However, this presents two important disadvantages, among others: on the one hand, the stiffness of titanium (Ti) is higher (100–110 GPa) than the cortical bone (20–25 GPa), which produces the stress-shielding phenomenon, promoting bone resorption surrounding the implant and compromising, in these cases, the functionality of the implants [11]; on the other hand, the inert biological character of titanium surfaces results in a poor cellular interaction between Ti and host bone tissue—an outcome that can affect the proper reconstruction of bone, resulting in implant loosening [12]. For these reasons, the development of Ti implants with satisfactory Young's modulus and best cellular interaction for bone tissue replacement remains a challenge to be addressed.

The design and manufacturing of materials with a Young's modulus close to that of cortical bone has been widely discussed in the scientific literature [13–16]. Among the possible solutions proposed is the use of porous materials [17–25]. Space holder techniques (SHT) have been identified as a low-cost route [26] to tailor porous titanium substrates with a high control of pore design (amount, morphology and pore size) which are key to obtain a balance response in relation to bone-implant integration and mechanical behavior. The large and irregular pores compromise the fatigue behavior of the implant as well as corrosion resistance, due to porosity [27,28]. However, at least a 100 μm average pore size [26,29–31] is required to guide the cellular response and synthesis bone matrix. This bone in growth through the pores also promotes long term stability and implant osseointegration. On the other hand, the inherent roughness to pore wall (increase of surface contact) and surface free energy, are both considered key factors for cell adhesion, migration and differentiation [32] while inhibiting the bacterial attachment and biofilm formation [33,34]. In this context, many reports have shown the interaction between roughness and cellular response and how the microstructure and nanostructure [35] can interfere and modulate cellular adhesion, spreading and differentiation of mesenchymal stem cells [36], osteoblasts [37,38], macrophages [39,40] and osteoclast [41]. The vast majority of surface modification treatments are based on the micro and submicron scale roughness development but recently the nanotopography architecture has shown an important role for guiding bone regeneration [35].

The need to characterize porosity is a fact. Different techniques have been used in the literature: Archimedes, helium pycnometry, micro-tomography, image analysis [42–45] and electrical impedance spectroscopy [35,46–48]. Some preliminary studies were developed using the latter technique to correlate porosity of titanium, pore size and electrical conductivity, although the influence of the electric frequency in the measured impedance was not described [35]. Furthermore, in this work, the electrochemical impedance spectroscopy is used to conclude that the pores in the porous titanium play a negative part in corrosion resistance and the flowing electrolyte can increase the corrosive rate of all titanium samples. However, the influence of pore morphology on the corrosion of porous titanium implants was previously studied in [48]. In this study, small and isolated pores seemed to encourage the stagnation of electrolyte, preventing its free flow crucial to the ion incorporation/titanium release process of passivation [48]. On another hand, the oxide layer formed in in-vivo conditions on the titanium surface has also been studied in [46,47], and an electrical model was proposed, to characterize the corrosion processes occurring in in-vivo situations.

Traditionally, the techniques that are required to evaluate and monitor the renewal, differentiation and maturation of osteoblast cells in culture after different experimental conditions are generally destructive techniques, such as histology, scanning electron microscopy, fluorescence microscopy, immunohistochemistry and other biochemical assays. Recently, as an alternative approach,

impedance-based cellular assays in the field of tissue engineering and regenerative medicine offer a range of methods that use microelectrodes to measure the impedance of biological systems, and thus obtain information of cellular behavior growing on any surface or biomaterials [49]. In this work, it is studied and discussed the use of electrical impedance spectroscopy to characterize both the porosity size of porous titanium discs produced by space holder technique and the growth of osteoblast cell cultures on the flat surface and inside of pores.

## 2. Materials and Methods

Scheme 1 summarizes the manufacturing routes of titanium discs, as well as the characterization carried out in this work. It shows the use of the electrical impedance spectroscopy to assess the in-situ cellular behavior of osteoblast growing on porous discs.

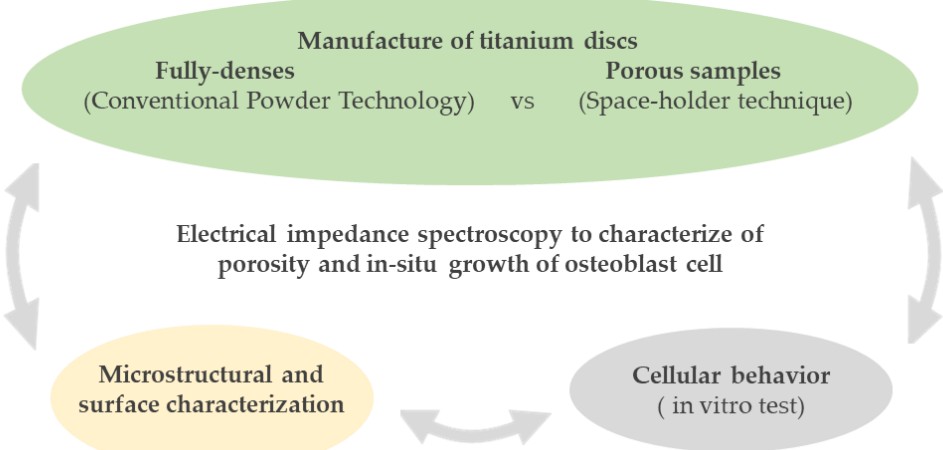

**Scheme 1.** Multidisciplinary approach followed for the in-vitro and in-situ characterization of cellular growth in the studied samples.

### 2.1. Manufacture of Titanium Discs

Fully-dense titanium samples were obtained by conventional Powder Metallurgy technology (PM), using initial metallic powder equivalent to cp Ti Grade IV (ASTM F67-00), produced by a hydrogenation/dehydrogenation process and supplied by SE-JONG Materials Co. Ltd., Incheon, Korea. Porous substrates were fabricated using the space-holder technique. The ammonium bicarbonate, $NH_4HCO_3$, (Cymit Química SL, Barcelona, Spain, with a minimum purity of 99.9%) was used as spacer. The 40% nominal volume of pores was selected, and two ranges of pore size (100–200 μm and 355–500 μm) were used. Next, the mixture of the titanium powder and the spacer is pressed to 800 MPa using a universal Instron machine to obtain green body samples (12 mm of diameter and 5 mm of height). Before sintering, the spacer was thermally removed (firstly at 60 °C and, then at 110 °C, carrying out both stages of the thermal treatment for 10 h and low vacuum conditions of $10^{-2}$ mbar). The fully-dense substrates were obtained by conventional powder metallurgy, used a compaction pressure of 1300 MPa and a sintering temperature of 1300 °C (same time and atmosphere of sintering above).

### 2.2. Porosity Characterization

Different techniques were employed to characterize the porosity at different levels (superficial and volumetric porosities). Density measurement was performed through the Archimedes' method (ASTM C373-88). Total ($P_T$) and interconnected porosities ($Pi$) have been determined from these measurements. On the other hand, the image analysis (IA) was carried out using a Nikon Epiphot optical microscope coupled with a Jenoptik Progres C3 camera and Image-Pro Plus 6.2 analysis software.

In addition, to contrast the total porosity values on the surface, this technique allowed us to measure the equivalent diameter ($D_{eq}$), defined as the average diameter measured from the pore centroid and the pore shape factor, $F_f = 4\pi A/(PE)^2$, where A is the pore area and $PE$ is the experimental perimeter of the pore. Furthermore, the characterization of roughness in the inside of pores was carried out by both high-resolution scanning electron microscopy, SEM (FEI Teneo, FEI, Eindhoven, The Netherlands) and confocal-laser microscopy, CLM (SENSOFAR S Neox, Leica, Glonn, Germany). In this study, the arithmetical mean deviation (Sa) is evaluated (ISO 25178).

Finally, electrical impedance spectroscopy technique [50] was used initially to assess and corroborate the porosity obtained and later to characterize osteoblast cell culture growth on the porous titanium discs (porous and fully-dense; see details in the Section 2.3.5). The equipment used to perform the electrical impedance measurements was the Hewlett-Packard 4395A, which is a network, spectrum and impedance analyzer, available at IMSE-CNM-CSIC (Instituto de Microelectrónica de Sevilla, Seville, Spain). An initial calibration procedure for the Hewlett-Packard impedance analyzer was carried out, using the fabricant calibration impedances. The impedance modulus of the titanium samples (fully-dense and 40% volumetric porosity, with the different pore sizes 100–200 μm and 355–500 μm) was measured at the available frequency range from 150 kHz to 500 MHz.

## 2.3. In-Vitro Cell Experiments

MC3T3E1 has been used to evaluate the porosity's effect on cell metabolism and viability during cell proliferation and differentiation process and correlate these parameters with the bioimpedance results.

### 2.3.1. Cell Culture

MC3T3E1, a murine pre-osteoblast cell line (CRL-2593, from ATCC, Manassas, VA, USA) was used to analyze the porosity effect on cell metabolism and viability during cell adhesion and proliferation process. Routine passaging of the cell line was performed on 25 cm² flasks with Minimum Essential Medium (MEM), containing 10% fetal bovine serum plus antibiotics (100 U/mL penicillin and 100 mg/mL streptomycin sulfate) (Invitrogen). Porous titanium discs were sterilized by autoclave at 121 °C for 30 min and then were placed into a 24-well plate. Osteoblasts cells were seeded at a cellular density of 30,000 cells/cm² per sample and culture in 800 μL of prewarmed culture medium. Plates were kept at 37 °C in a humidified 5% CO2 atmosphere and as a controls, triplicate blank, TCP (Tissue culture plastic) and fully-dense pure titanium disc were used as negative and positive controls in the same plate for each time period.

At 48 h of osteoblast cultured on porous Ti samples, cell media was changed to osteogenic media (α-MEM medium) supplemented with 10 mM ascorbic acid (Merck, Germany) and 50 μg/mL of β-glycerophosphate (StemCell Technologies, Canada). In-vitro experiments were carried out at 4, 7 and 21 days of cell incubation in which the samples were transferred to a new 24-well plate to avoid counting non-attached or attached cells on the well plate.

### 2.3.2. Cell Viability and Proliferation Assay

Cell viability and proliferation tests were evaluated using AlamarBlue® reagent (Invitrogen, Waltham, MA, USA). According to manufacturer's protocol, new fresh media (800 μL) and 80 μL of AlamarBlue® reagent were added and the plate was incubated during 1 h 30 min at 37 °C in dark conditions. Later the absorbance at 570 nm (TECAN, Infinity 200 Pro) was recorded.

### 2.3.3. Cell Differentiation by Alkaline Phosphatase (ALP) Activity

MC3T3 differentiation levels were evaluated through alkaline phosphatase (ALP) activity, using the Alkaline Phosphatase Assay kit Colorimetric (Abcam ab83369, Abcam, Cambridge, UK). The assay was performed on days 4, 7 and 21 by triplicate according to manufacturer's protocol. The absorbance at 405 nm of 4-nitrophenol was measured in a 96-well microplate reader. Data were expressed as μmol/min/mL of pNPP (para-Nitrophenylphosphate).

### 2.3.4. Cell Morphology

Scanning electron microscopy (SEM) was used to evaluate the cell morphology at 4, 14 and 21 days. The samples were fixed in 10% formalin followed by a dehydration step with ethanolic solutions and coated by gold-coating using a sputter coater (Pelco 91000, Ted Pella, Redding, CA, USA). The images were obtained using a Jeol JSM-6330F scanning electron microscope (JEOL, Tokyo, Japan) with an acceleration voltage of 10 kV.

### 2.3.5. Electrical Impedance of Osteoblast Growth

Studied specimens were placed into 24-well plate and seeded with osteoblast at a cell density of 30,000 cells/cm$^2$. Using the same equipment (Hewlett-Packard) previously described in the characterization Section 2.2, and after the initial calibration, impedance modulus measurements were taken at 250 MHz frequency at 4, and 14 days of cell culture. During the impedance experiments we used as negative controls the same specimens, placed in same cell media (Minimum Essential Medium, as explained in Section 2.3.1) but without cells.

### 2.4. Statistical Analysis

All experiments were performed in triplicate in order to ensure reproducibility. Results were expressed as mean and standard deviation to perform two-way ANOVA followed by a Tukey's post-test using OriginPro 2019 software (OriginLab, Northampton, MA, USA) for Windows Significance level was considered at $p$-values of $p < 0.05$ (*) and $p < 0.01$ (**).

## 3. Results and Discussion

In the following sections, the main results and the discussion related to the microstructural characterization and the cellular behavior of the manufactured porous discs are presented. The potentialities of using the electrical bioimpedance measurement technique to discriminate the type of porosity, as well as the presence of osteoblasts attached to the surfaces of the titanium disks, are described.

### 3.1. Characterization of the Porosity of Titanium Samples

Figure 1 and Table 1 show the comparison of the porosity obtained for the three studied substrates: fully-dense as a reference and two designed with a 40 vol.% of spacer and two differences size ranges, manufactured according to the conditions described in Section 2.1. First, the macrographs shown in Figure 1A reveal a micro and macropores distribution homogenous on the titanium samples. Figure 1B,C show the microstructure of the titanium substrates and allow the evaluating of the porosity features, in terms of content, size and morphology. These features are consistent with the values of total porosity and equivalent diameter, in both ranges of spacer sizes studied (see Table 1). Finally, Figure 1D,E show the 3D topography of the surface of the samples. In Table 1 it summarizes the characterization values performed by the Archimedes method and by IA techniques. These techniques allow the determination of the porous density value, total and interconnected porosity, as well as the equivalent diameter of pores and the pore shape factor while the roughness characterization of inside of pores and on flat surface (arithmetical mean deviation) was obtained with the confocal-laser microscopy technique.

The micro-porosity in the fully-dense was residual (dense 4.37 g/cm$^3$, versus Ti by cast 4.5 g/cm$^3$ [51]) and presented a small size and rounded morphology, as expected. On the other hand, it is observed that the contents (40.2 and 40.8 vol.%) and the average sizes (226 and 359 μm) of the pores obtained experimentally, are consistent with the content and size range of the spacers used, corroborating the potentialities of the space holder technique in terms of cost, reliability, reproducibility and potential use on an industrial scale (incorporating the spacer removal route in the conventional process). The obtained macropores with the spacer have a more irregular morphology than those inherent in the

sintering stage. The properties of these porous titanium substrates (content, size and morphology), as well as the quality of the sintering necks, are responsible for the mechanical and fatigue resistance of the porous materials ($E_d$ = 117.9, 60.6 and 63.2 GPa for fully-dense and a 40 vol.% of spacer of two differences size ranges, respectively). In this context, a range of spacers between 100–200 μm is recommended to guarantee at the same time the biomechanical requirements of the cortical bone tissue (150–180 MPa [52]) and allow its growth through the interconnected porosity (>100 μm [53]). In addition, the micro roughness of the pores walls depends on their size. In general, the roughness obtained is adequate to enhance the adhesion of osteoblasts (see discussion details below).

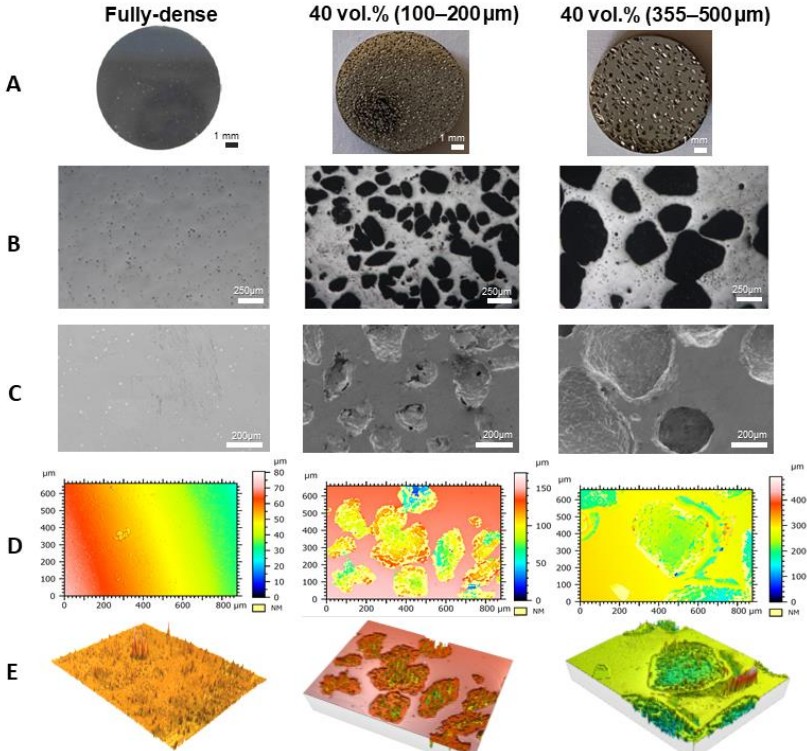

**Figure 1.** Image and roughness analysis of titanium substrates: fully-dense and with a 40 vol.% of spacer for two range size (100–200 μm and 355–500 μm). Figure (**A**), (**B**) and (**C**): Macrographs (high resolution camera), micrographs (Image Analysis) and SEM, respectively. Figure (**D**) and (**E**): porosity reconstruction performed by Sensomap 6.2 software (Leica, Glonn, Germany) (confocal-laser microscopy) and 3D topography of the surface of the samples.

**Table 1.** Density, porosity and roughness evaluation obtained by Archimedes method, image analysis and confocal-laser microscopy, respectively.

| Spacer | | Archimedes Method | | | Image Analysis | | | Confocal-Laser, $Sa$ (μm) | |
|---|---|---|---|---|---|---|---|---|---|
| Volumen (%) | Size Range (μm) | Density (g/cm³) | $P_T$ (%) | $P_i$ (%) | $P_T$ (%) | $D_{eq}$ (μm) | $F_f$ | | |
| 0 | - | 4.37 ± 0.02 | 2.4 ± 0.3 | 1.3 ± 0.3 | 1.8 ± 1.2 | 4.6 ± 1.0 | 0.96 ± 0.11 | All surface | 0.7 |
| 40 | 100–200 | 2.69 ± 0.02 | 40.2 ± 0.6 | 32.9 ± 0.8 | 41.6± 3.3 | 226 ± 178 | 0.60 ± 0.2 | Flat area between the pores | 5.0 |
| | | | | | | | | All surface | 15.3 |
| | 355–500 | 2.67 ± 0.02 | 40.8 ± 0.5 | 27.9 ± 0.7 | 43.4 ± 7.8 | 359 ± 223 | 0.71 ± 0.2 | Flat area between the pores | 3.6 |
| | | | | | | | | All surface | 31.8 |

The mean impedance values measured for the different porous discs, at different frequencies are shown in Figure 2. We can observe that electrical impedance can be a suitable marker of the porosity of the analyzed samples. The impedance results show the different pore size used in this study. At higher frequencies around 500 MHz, the differences on pore size are more significant.

The resistance and conductance can be frequency-dependent, a phenomenon known as the universal dielectric response [52]. We can observe the approximated linear response obtained in the modulus of impedance in the different samples in Figure 2A. Differences in the slope (higher with increasing pore size) can be better understood if we take into account the normalized area of the samples. If we normalize the modulus of the impedance (Figure 2B), taking as a reference the fully-dense samples, we can observe an almost parallel distribution of the impedance modulus for each of the volumetric porosities.

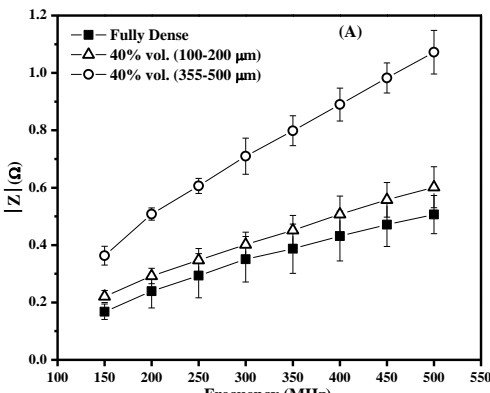 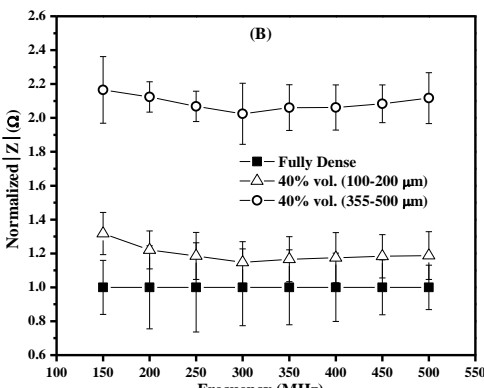

**Figure 2.** (**A**) Modulus of impedance, |Z|, and (**B**) normalized modulus of impedance, |Z| vs. frequency, measured for all specimens without cells and media.

### 3.2. Biological Response of Osteoblasts

Ti samples with 40 vol.% porosity and different pore size are tested to assess how surface modification affects their biological properties. Thus, we plated osteoblasts (MC3T3 cell line) on fully-dense and porous discs to check their cellular behavior. We tested the viability, proliferation and differentiation through cell metabolic activity, Alkaline phosphatase assay and their cell morphology by SEM at different time points.

### 3.2.1. Cell Viability

The evaluation of cell viability at 24 h was tested on seeded porous Ti scaffolds compiling the results in Figure 3. We could observe viable osteoblasts and in attached stage due to the detection of higher levels of osteoblast cell metabolism. The substrates reached more than 80% of cell viability confirming the safety porous route and the biocompatibility of the technique. We observed significant differences among fully-dense and 40 vol.% porosity, in both porous sizes.

In Figure 4 we evaluated the osteoblast cell viability at long-term, particularly at 4, 7 and 21 days of cell incubation. At 21 days, all surfaces showed more than 80% of metabolic activity which also denotes a suitable cell adhesion and proliferation in-vitro response. We could observe statistics differences of metabolic activity levels on porous 40 vol.% (100–200 μm) (*p*-value < 0.05). The porosity increases the surface contact area which promotes higher osteoblast cell density attached on the surface compared to fully-dense. Additionally, 40 vol.% (100–200 μm) porous sample showed higher roughness values (5 μm) that also favors cell adhesion.

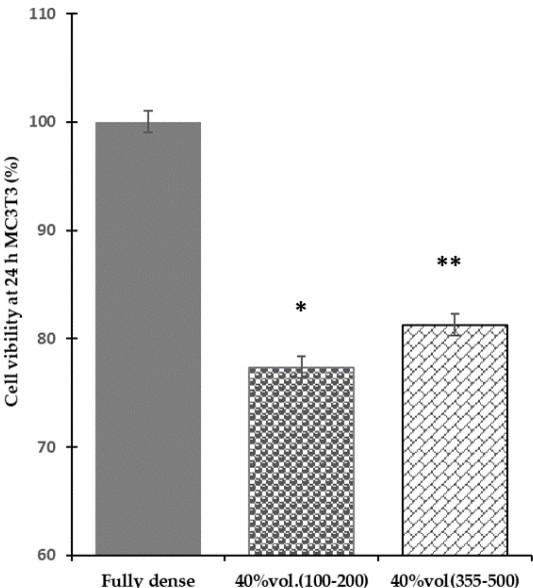

**Figure 3.** Cell metabolic activity in adhesion stage determined by AlamarBlue assay of MC3T3 cell line at 24 h. (* $p < 0.05$ fully-dense vs. 40 vol.% (100–200 μm) and ** $p < 0.001$ fully-dense vs. 40 vol.% (355–500 μm)).

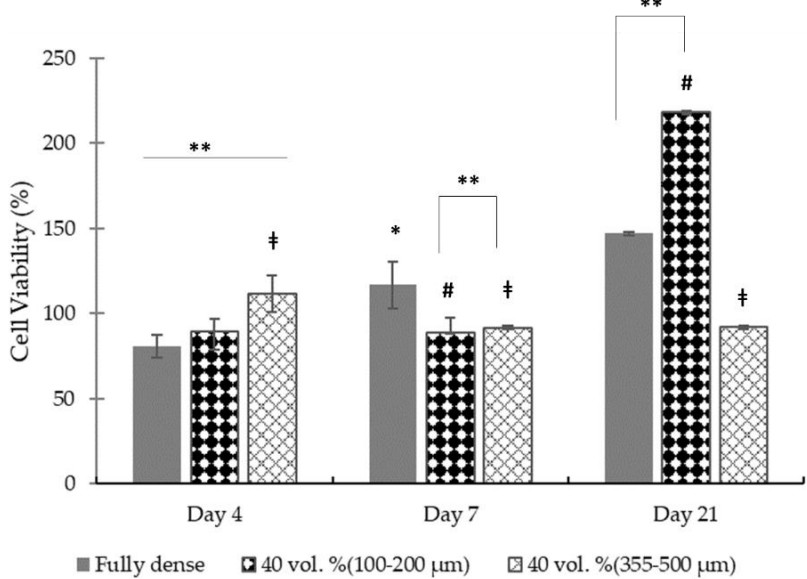

**Figure 4.** Cell metabolic activity in proliferation stage determined by AlamarBlue assay of MC3T3 cell line at 4, 7 and 21 days. Significance level at $p$-value < 0.05; (*) fully-dense 4 days vs. 21 and 7 days vs. 21 days. (#) 40 vol.% (100–200 μm) 4 days vs. 7 days and 21 days; (‡) 40 vol.% (355–500 μm) 4 days vs. 7 and 21 days, 7 days vs. 21 days. A $p$-value of <0.01 (**) indicate statistics differences.

### 3.2.2. Cell Differentiation by ALP Evaluation

In Figure 5, the ALP activity detected on 40 vol.% (100–200) μm was higher in comparison with fully-dense at all times of cell culture, especially at 21 days in which differences are statistically significant (* $p$-value of <0.05). These results highlight the optimal functionality of pre-osteoblast that are capable to differentiate into mature osteoblast greater on 40 vol.% (100–200) μm.

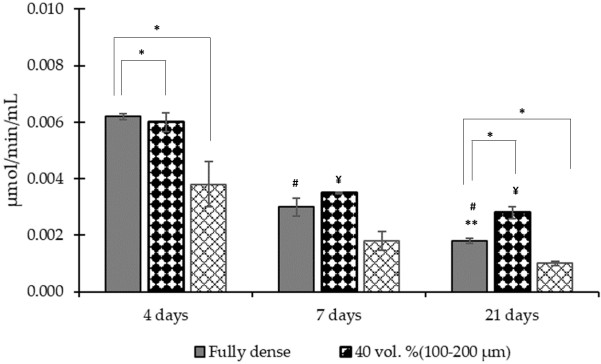

**Figure 5.** Alkaline phosphatase (ALP) quantification at 4, 7 and 21 days of MC3T3 growing on fully-dense surface, sample with porous 40 vol.% (100–200) μm and 40 vol.% (355–500) μm. A * $p$-value of < 0.05. # $p < 0.005$ fully-dense 4 days vs. 7 days and 21 days; ¥ $p < 0.05$ 40 vol.% (100–200) μm 4 days vs. 7 days and 21 days; ** $p < 0.005$ 40 vol.% (355–500) μm 4 days vs. 21 days.

### 3.2.3. Cell Morphology

To further investigate the viability and bifunctionality of osteoblast, SEM images were taken at 4, 14 and 21 days (see Figure 6). The SEM images at short-term stage showed similar cell density, fibroblastic phenotype morphology and coverage on the three samples corroborating the cell viability results observed in Figure 3. Furthermore, at day 14 and 21 osteoblast cells are totally covering pores on 40 vol.% (100–200 μm) which facilitates the higher contact between filopodia protrusions to surface and to neighboring cells. This also favors the matrix deposition on porous surfaces developing a mature ECM (extracellular matrix). Proof of this cellular behavior has been described in previous studies of our group using $NH_4HCO_3$ porous substrates using 50% of spacer particles [39,50]. At day 21, it can be observed a cell monolayer of highly differentiated osteoblast on 40 vol.% (100–200 μm) sample with deposition of hydroxyapatite vesicles which also corroborates the advanced mineralization stage of osteoblast in this porous surface. Small pore size also develops higher number of contact and interactions among osteoblast which is essential for bone matrix production and cell proliferation.

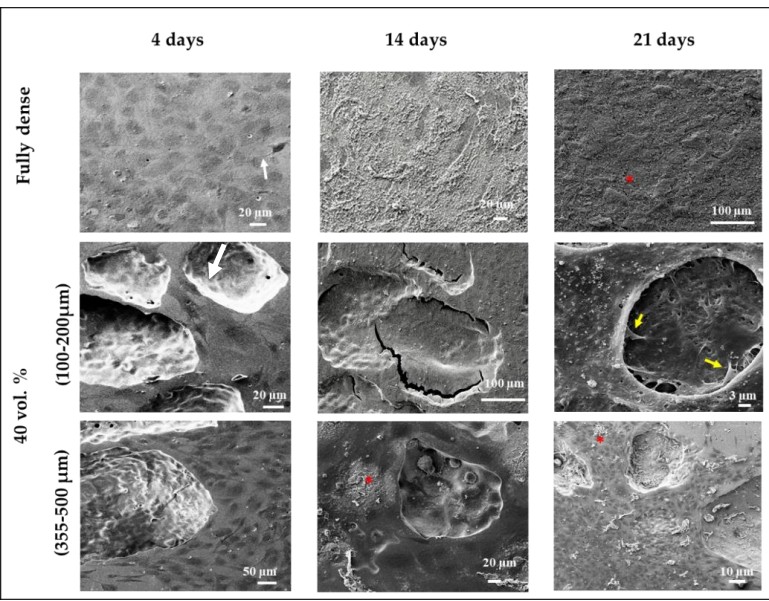

**Figure 6.** SEM micrographs of MC3T3 cell morphology at 4, 14 and 21 days of growing on the surface fully-dense sample, sample with porous 40 vol.% (100–200) μm and 40 vol.% (355–500) μm. Cell–cell interaction (white arrow), cell-surface junction (yellow arrows) and hydroxyapatite (red asterisks) are indicated in the image.

Other research studies have evaluated the in-vitro and in-vivo biological response of porous scaffolds highlighting the pore size and morphology as important factors for bone tissue repair. In that sense, small pores of around 100 µm favor a hypoxia condition which promotes osteochondral formation whereas bigger pores of 100 µm to 500 µm allow for direct osteogenesis. This fact has been related to the new vascularized system that is able to penetrate through the porous structure to irrigate the new bone formation [40].

Additionally, previous results in our group have reported that porous scaffolds of 150–300 µm offer higher mechanical strength and stability to anchor with adjacent tissue which improves implant osseointegration than 300 to 500 µm pore size [40].

### 3.2.4. Characterization of Osteoblasts Growth with Impedance Spectroscopy

The measured electrical impedance for the different samples after 4 days and 14 days of cells cultivation is shown in Figure 7. After 4 days of osteoblast cell culture, Figure 7 shows how the modulus of the electrical impedance measured increased for all specimens where cells were cultivated in comparison with the control specimens, placed in same cell media but with no seeded cells. This result suggests the cell growth and proliferation at 4 days produce a new resistance increasing the values on seeded specimens compared to their counterparts with no cells. This result was especially significant in the case of the fully-dense seeded with cells, where the increase of impedance was really abrupt. This phenomenon can be attributable to the deposition and growth of all cells in the external surface of the specimens, while in the porous substrates, the cells are deposited and growth in both the external surface and internal surface of the interconnected porosity. Thus, in the fully-dense specimen, the concentration of the cells in the external surface has to be undoubtedly higher than in the porous specimens at the same cultivation days. This was also confirmed by SEM images (Figure 6) in which fully-dense sample showed a cellular monolayer covering the entire surface while porous substrates showed cells inside the pores and on flat surfaces as well. This fact highlights the importance of surface properties in the measured impedance of the sample.

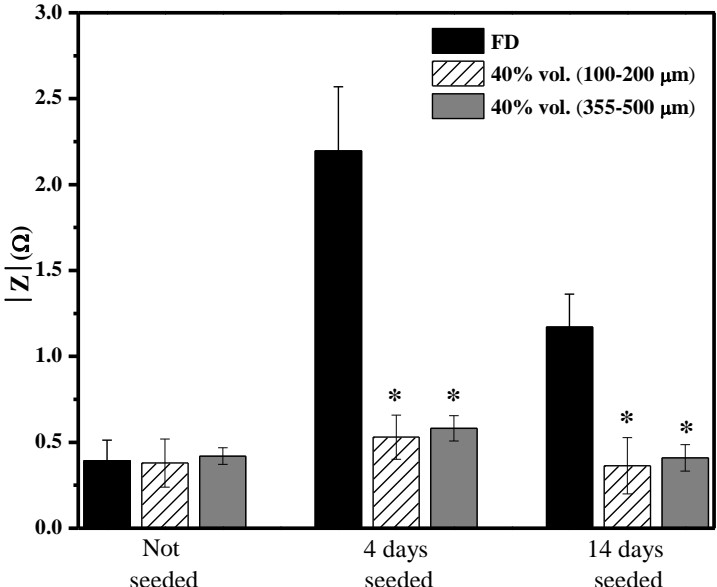

**Figure 7.** Modulus of electrical impedance (|Z|) measured after 4, and 14 days in fully-dense samples (FD), 40 vol.% (100–200 µm) and 40 vol.% (355–500 µm). (*) Significance level at *p*-value < 0.05 FD vs. 40 vol.% (100–200 µm) or 40 vol.% (355–500 µm).

However, at 14 days of cell culture the impedance decreased for all samples with cells (fully-dense, 40 vol.% (100–200 µm) and 40 vol.% (355–500 µm). The reduction in the measured impedance is not related to cell death since, as the in-vitro experiments demonstrate, the 14-day cell culture shows a high

percentage of viability in all the samples tested and the cells show ALP activity. Nordberg et al. [54] described a first phase of increasing impedance, coinciding with the proliferation of the osteogenic cell culture, which we could observed it at 4-day measurement, followed by a "drop-phase" that coincides with the initial production of extracellular matrix deposition by mature osteoblasts. This fact indicates that osteoblasts are in the late differentiation phase, and early mineralization stage which also is correlated by a less dense cell coverage and a fewer cell to cell junctions. In the SEM images we can see how the cells at 14 days begin to deposit molecules with the appearance of calcium carbonate and with the typical hexagonal structure of hydroxyapatite following the characteristic pattern of osteogenic differentiation.

## 4. Conclusions

In this work we explore the use of impedance spectroscopy to characterize the porosity of different titanium samples, and to correlate cell response of osteoblast growing in-situ on porous titanium samples, as a potential technique for the real-time measurement of osseointegration.

The differences of impedance measurements between samples were more significant at higher frequencies (around 500 MHz). On the other hand, osteoblasts cells (MC3T3 cell line) showed 80% of cell viability at 24 h and higher cell metabolic activity was observed on porous titanium surfaces with a higher cell adhesion and proliferation when compared to fully-dense samples. An increase impedance measurement was observed for all titanium samples where cells were cultivated. This increase was especially significant in the case of the fully-dense samples (with an approximated 640% of increase) which can be explained by the resistance of attached cells on samples surface. However, at 14 days of cell culture a drop phase was detected (around 50%), in line with previous published works. In conclusion, electrical impedance can be used a reliable marker of the porosity of titanium implants facilitating the characterization of pore parameters on different substrates. Further work is needed to explain the reasons for this impedance decrease and correctly make use of these bioimpedance measurements for cell culture monitoring.

**Author Contributions:** Conceptualization, project administration, supervision, methodology, M.-J.M.-G., and Y.T. Investigation, formal analysis, validation, M.G., A.O., P.T., E.C., A.C., M.Á.V. and M.H. Discussion and writing—original draft preparation, all the authors. All authors have read and agreed to the published version of the manuscript.

**Funding:** This work was supported by the Ministry of Science and Innovation of Spain under the grant PID2019-109371GB-I00, by the Junta de Andalucía–FEDER (Spain) through the Project Ref. US-1259771 and by the Junta de Andalucía-Proyecto de Excelencia (Spain) P18-FR-2038.

**Conflicts of Interest:** The authors declare no conflict of interest.

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
