# Peer review of "Use of Impedance Spectroscopy for the Characterization of In-Vitro Osteoblast Cell Response in Porous Titanium Bone Implants"

_metals, doi:10.3390/met10081077_

Round 1
Reviewer 1 Report
Cellular titanium alloys are very important for biomedical applications. Assessment of the cellular response of implants are relatively very expensive, long and delicate. This work proposed the use of impedance spectroscopy to characterize osteoblast growth in cell cultures on porous implants. This is interesting and has potential for publication in the journal. However, revisions are required to further improve the manuscript quality by addressing the following comments.
- Abstract contains too much background. Please make it concise and contain more results.
2. Introduction: it is quite well known that additive manufacturing (selective lase melting and electron beam melting) can perfectly manufacture porous titanium alloys for biomedical titanium alloys and the produced porous titanium alloys demonstrate very good fatigue properties. However, such references were missing. Some representative references should be added, such as Acta Materialia 126 (2017) 58-66; International Journal of Fatigue 119 (2019) 173-184; Acta Materialia 113 (2016) 56-67.
3. Introduction: it is highly to add work on the characterization of the corrosion behavior of porous titanium alloys, such as Journal of Bio- and Tribo-Corrosion 5, (2019) 3.
- Writing skills should be further improved, e.g. (1) verbs in past tense should be used to describe the experiments; (2) the multiple references in one location should be combined, i.e. [xx-xx].
- When working out the Electrical resistance (real part of the impedance) using the Equation (1), how did you use the area of the samples? Note that the area was not that accurate because of the inhomogeneous distribution of porous/cellular structure in the sample.
- Results should be discussed in terms of the microstructure and the distribution of porous structures in the 3 types of samples.
Author Response
Dear reviewer,
We would like to thank you for giving us the opportunity to submit to Metals the revised version of the manuscript entitled: “Use of Impedance Spectroscopy for the characterization of in vitro osteoblast cell response in porous titanium bone implants ” by Mercè Giner, Alberto Olmo, Miguel Hernández, Paloma Trueba , Ernesto Chicardi, Ana Civantos, María Ángeles Vázquez , María-José Montoya-García and Yadir Torres.
We would also like to thank the reviewers for providing insightful and important comments on our manuscript. As requested, we have considered the reviewers’ comments (R.C.), answered with our list of responses (A.R.). We have highlighted all changes in the text, clarifying comments in blue. In addition, the text has been thoroughly revised for typographical errors as well as errors in syntax.
Reviewer #1:
Cellular titanium alloys are very important for biomedical applications. Assessment of the cellular response of implants are relatively very expensive, long and delicate. This work proposed the use of impedance spectroscopy to characterize osteoblast growth in cell cultures on porous implants. This is interesting and has potential for publication in the journal. However, revisions are required to further improve the manuscript quality by addressing the following comments.
R.C. Abstract contains too much background. Please make it concise and contain more results.
A.R. Thank you very much for your comment, we have rewritten the abstract to try to improve the aspects that you suggest.
R.C. Introduction: it is quite well known that additive manufacturing (selective lase melting and electron beam melting) can perfectly manufacture porous titanium alloys for biomedical titanium alloys and the produced porous titanium alloys demonstrate very good fatigue properties. However, such references were missing. Some representative references should be added, such as Acta Materialia 126 (2017) 58-66; International Journal of Fatigue 119 (2019) 173-184; Acta Materialia 113 (2016) 56-67.
A.R. Thank you very much for your comment, we agree with you. Laser structuring is a promising technique that presents advantages over other conventional structuring techniques but it is very expensive nowadays. In that sense, we have introduced the references that you suggest in the manuscript.
Acta Materialia 126 (2017) 58-66
International Journal of Fatigue 119 (2019) 173-184
R.C. Introduction: it is highly to add work on the characterization of the corrosion behavior of porous titanium alloys, such as Journal of Bio- and Tribo-Corrosion 5, (2019) 3.
A.R. We have rewritten this topic line and we have introduced references acording to your suggestion in the manuscript.
Journal of the Mechanical Behavior of Biomedical Materials 69 (2017) 144-152
Corrosion Science 156 (2019) 106-116
R.C.Writing skills should be further improved, e.g. (1) verbs in past tense should be used to describe the experiments; (2) the multiple references in one location should be combined, i.e. [xx-xx].
A.R. We have rewritten the manuscript sorting out these issues.
R.C. When working out the Electrical resistance (real part of the impedance) using the Equation (1), how did you use the area of the samples? Note that the area was not that accurate because of the inhomogeneous distribution of porous/cellular structure in the sample.
A.R. Many thanks for the comment. We have tried to clarify in a better way our manuscript taking into account your question. What we measured with our equipment (Hewlett-Packard 4395A) was the modulus of the impedance, as it is now better defined in sections 2.2 and 2.3.5. As the real part of the impedance was not used in the characterization of the porous titanium samples or cell cultures, we think it is better to remove equation 1, in order to avoid possible misunderstandings.
Also, as commented in the manuscript text, we show in fig 2 (A) the modulus of impedance, |Z|, and in (B) the normalized modulus of impedance, (dividing by the impedance of fully dense samples), vs frequency, obtaining this way information independent from the total area of the sample. As can be seen and as it is commented in the conclusions section, electrical impedance can be a good marker of the porosity of the analyzed samples, being able to characterize and differentiate the size of pore used (in a more precise way as with other techniques, such as the Arquimedes method or the Image Analysis (table 1)).
R.C.Results should be discussed in terms of the microstructure and the distribution of porous structures in the 3 types of samples.
A.R. Thank you very much for the reviewer's suggestion. The authors have reviewed the discussion of the results related to the influence of porosity. We think that these are consistent with the tests carried out in this work. In this context, although other small comments could be made, they would not cease to be speculative, we should do other types of additional tests, which we are planning for other future work.
Thank you for your consideration of our work.
Sincerely,
Mercè Giner, PhD

Reviewer 2 Report
There are some comments:
- There does not indicate the chemical composition of titanium Grade 4 in the methods. The titanium Grade 4 may contain different amounts of Fe. The amount of impurities can affect chemical and biological processes. The information about purity of Grade 4 will useful for comparing your results with other studies.
- Typo on lines 142, 144, 188 (Hewkett-Packard)
- Why is the roughness estimated by the Sa parameter? Usually the roughness is estimated by the parameter Ra (the arithmetical mean deviation of the profile)
Author Response
Dear reviewer,
We would like to thank you for giving us the opportunity to submit to Metals the revised version of the manuscript entitled: “Use of Impedance Spectroscopy for the characterization of in vitro osteoblast cell response in porous titanium bone implants ” by Mercè Giner, Alberto Olmo, Miguel Hernández, Paloma Trueba , Ernesto Chicardi, Ana Civantos, María Ángeles Vázquez , María-José Montoya-García and Yadir Torres.
We would also like to thank the reviewers for providing insightful and important comments on our manuscript. As requested, we have considered the reviewers’ comments (R.C.), answered with our list of responses (A.R.). We have highlighted all changes in the text, clarifying comments in blue. In addition, the text has been thoroughly revised for typographical errors as well as errors in syntax.
Reviewer #2:
There are some comments:
R.C. There does not indicate the chemical composition of titanium Grade 4 in the methods. The titanium Grade 4 may contain different amounts of Fe. The amount of impurities can affect chemical and biological processes. The information about purity of Grade 4 will useful for comparing your results with other studies.
A.R. Thank you very much for the comment. The composition of comercial pure titanium is acording with ASTM F67-00 for Grade 4 (line 117).
R.C. Typo on lines 142, 144, 188 (Hewkett-Packard)
A.R. We have corrected the mistake and We have eliminated the model of the impedance analyzer from lines 144 and 188.
R.C. Why is the roughness estimated by the Sa parameter? Usually the roughness is estimated by the parameter Ra (the arithmetical mean deviation of the profile)
A.R. In this study, the arithmetical mean deviation (Sa) and the root mean square height (Sq) were evaluated (ISO 25178), using the images obtained with confocal-laser microscopy, CLM, (SENSOSAR S Neox, Leica, Germany). This measure of the surface roughness is more representative. The values that the reviewer comments, Ra, Rq, Rz and Ry, are inherent in a conventional profile obtained with a profilometer, following a line on the surface to be evaluated.
Thank you for your consideration of our work.
Sincerely,
Mercè Giner, PhD

Round 2
Reviewer 1 Report
The revision is acceptable for publication.